# Influence of Two Hexose Transporters on Substrate Affinity and Pathogenicity in *Magnaporthe oryzae*

**DOI:** 10.3390/microorganisms12040681

**Published:** 2024-03-28

**Authors:** Tinghong Huang, Dekang Guo, Xiao Luo, Ronghua Chen, Wenjuan Wang, Hanhong Xu, Shen Chen, Fei Lin

**Affiliations:** 1National Key Laboratory of Green Pesticide/Key Laboratory of Natural Pesticide and Chemical Biology, Ministry of Education, South China Agricultural University, Guangzhou 510642, China; 2Guangdong Provincial Key Laboratory of High Technology for Plant Protection, Guangzhou 510642, China

**Keywords:** hexose transporters, pathogenicity, substrate affinity, *Magnaporthe oryzae*

## Abstract

Hexose transporters (HXT) play a crucial role in the pathogenicity of *Magnaporthe oryzae*, serving not only as key facilitators for acquiring and transporting sugar nutrients to support pathogen development, but also as sugar sensors which receive transduction signals. The objective of this study is to investigate the impact of *MoHXT1-3* on rice pathogenicity and hexose affinity. *MoHXT1-3* deletion mutants were generated using CRISPR/Cas9 technology, and their affinity for hexose was evaluated through yeast complementation assays and electrophysiological experiments in *Xenopus oocytes*. The results suggest that *MoHXT1* does not contribute to melanin formation or hexose transportation processes. Conversely, *MoHXT2*, despite displaying lower affinity towards the hexoses tested in comparison to *MoHXT3*, is likely to have a more substantial impact on pathogenicity. The analysis of the transcription profiles demonstrated that the deletion of *MoHXT2* caused a decrease in the expression of *MoHXT3*, whereas the knockout of *MoHXT3* resulted in an upregulation of *MoHXT2* transcription. It is noteworthy that the *MoHXT2^M145K^* variant displayed an incapacity to transport hexoses. This investigation into the functional differences in hexose transporters in *Magnaporthe oryzae* provides insights into potential advances in new strategies to target hexose transporters to combat rice blast by blocking carbon nutrient supply.

## 1. Introduction

Pathogens involved in plant interactions must acquire metabolites from their hosts in order to fulfill their nutritional needs for growth. Sugar, being the primary carbon source, not only supplies energy for pathogen growth and plant defense responses, but also functions as a signaling molecule in pathogen–plant interactions [1]. Due to the limited ability of fungal pathogens to directly uptake sucrose from plants, hexose, primarily glucose, serves as the predominant sugar source obtained by pathogens from extracellular host cells. Consequently, sucrose present in plant cell walls undergoes hydrolysis into hexose through the action of invertase (INV), which is secreted by pathogenic fungi. Subsequently, hexose transporters (HXTs) facilitate the uptake and transfer of hexose. Notably, the cooperative functioning of INV and HXTs plays a pivotal role in the pathogenicity of fungi, irrespective of whether they are biotrophic or necrotrophic/hemibiotrophic pathogens [2]. An instance of this can be observed in the biotrophic rust fungus *Uromyces fabae*, where the genes *INV1* and *HXT1* are concurrently expressed within the haustoria during the infection phase. This suggests that rust fungi employ the INV enzyme to convert sucrose into monosaccharides, subsequently utilizing hexose-transporter transport mechanisms to acquire nutrition [3]. *Puccinia striiformis* f. sp. tritici, the pathogen responsible for wheat stripe rust, releases a significant quantity of INV upon infiltrating the host cytoplasm. In contrast, *PsHXT1* facilitates the transportation of hexose to sustain pathogen development. Through host-mediated silencing of the *PsHXT1* gene, it has been demonstrated for the first time that sugar deprivation impacts the in vivo growth and virulence of pathogens. This finding suggests that impeding sugar uptake by pathogens could be a potentially effective strategy for disease management by suppressing their proliferation and growth [4]. The genome of the hemibiotrophic fungus, *Colletotrichum graminicola*, harbors five *HXTs* (*CgHXT1–5*). Notably, *CgHXT1* and *CgHXT3* exhibit elevated transcript levels during the biotrophic infection stage, while *CgHXT2* and *CgHXT5* demonstrate increased expression at the necrotrophic stage [5]. An exception to this phenomenon is observed in the biotrophic fungus *Ustilago maydis*, which possesses a high-affinity sucrose transporter that competes with the host organism, thereby bypassing the need for sucrose degradation into hexose through the assistance of INV [6]. 

The uptake of extracellular monosaccharides into the cell is usually mediated by a carrier-mediated, facilitated diffusion [7]. It has been proposed that transport across the plasma membrane is regulated by an obligatory and complex sensor system, and at least 20 different hexose transporter-related proteins have been identified in baker’s yeast (*Saccharmoyces cerevisiae*) [8,9]. Among them, *HXT1*, *HXT2*, *HXT3*, *HXT6*, *HXT7* and *Gal2* are particularly important, as *hxt1-7*-deficient mutants are unable to grow in glucose, fructose, mannitol and sucrose media [10]. Catabolic hexose transporters demonstrate varying affinities towards their substrates, and the regulation of their respective genes is governed by glucose sensors in response to the presence or absence of carbon sources [9]. In addition, the roles of hexose transporters in glucose source uptake and regulation have been studied in *Neurospora crassa* and *Aspergillus nidulans* [11,12]. 

*Magnaporthe oryzae* is considered one of the foremost fungal pathogens due to its remarkable capacity to infect over 50 species of gramineous plants. It serves as a valuable model for studying interactions between plant hosts and pathogens. The hemibiotrophic lifecycle of rice blast fungi commences with a biotrophic primary invasion, encompassing spore germination, germ tube formation, appressorium colonization and host plant nail plate invasion, commonly referred to as the biotrophic phase [13]. During the subsequent necrotrophic phase, the plant cells undergo mortality as the rapidly expanding slender secondary hyphae penetrate the plant plasma membrane and proliferate within the host tissue [14]. Hence, in order to effectively respond to the nutrients provided by the host and acquire nourishment, fungi must develop intricate and efficient mechanisms for perceiving, acquiring and utilizing nutrients [15,16]. Experimental evidence has demonstrated that sugar transporters are specifically expressed during the initial stage of *M. oryzae* infection. Among these transporters, the hexose transporter *MoST1* plays a crucial role in the melanization of conidia and attachment spores. This finding suggests that the transfer and transportation of photosynthetic carbohydrate nutrients are also involved in the recognition and interaction between rice and rice blast. However, despite their close relation to *MoST1*, the three *HXTs* (*MoHXT1-3*) are unable to restore the pathogenicity of the *most1* mutant, indicating a functional divergence of the *HXTs* within this clade [17,18]. The assessment of monosaccharide transporter function in vivo is difficult due to the functional redundancy within the gene family. Electrophysiological experiments conducted in *Xenopus oocytes* have been widely utilized to express individual transporters heterologously and assess their substrate affinity by observing alterations in membrane potential, thus mitigating interference from redundant genes. We hereby present a comprehensive functional analysis of the three *MoHXTs* transporters in *M. oryzea*, encompassing both in vivo and in vitro investigations.

## 2. Materials and Methods

### 2.1. Strains, Plasmids, Media and Culturing Methods

The strains and plasmids utilized in this study are documented in Appendix A. The wild-type strain of *M. oryzae* (A60) and the HDR-deficient mutants Δ*Mohxt2* and Δ*Mohxt3* were cultured and maintained on complete medium (CM) agar plates at a temperature of 28 °C. Formulation of the CM was based on previous studies [19]. To culture mycelia from 5-day-old colonies in a liquid medium, they were transferred to liquid CM and subjected to shaking at a rate of 160 rpm at room temperature for a period of 36 h. The *S. cerevisiae* sugar transporter knockout strain *EBY.VW4000* was employed for in vivo complementation phenotype assays [8]. The yeast strains were cultured at a temperature of 30 °C in a synthetic medium (SC; 0.67% yeast nitrogen base without amino acids) that was enriched with either maltose or a distinct carbon source.

### 2.2. Knockout MoHXT1, MoHXT2 and MoHXT3 Using CRISPR-Cas9

In order to ascertain the roles of the sugar transporter genes *MoHXT1, MoHXT2* and *MoHXT3* in *M. oryzea*, we employed the CRISPR-Cas9 gene-editing technique to transiently introduce purified Cas9 protein and in vitro synthesized sgRNA into protoplasts of *M. grisea*. This approach allowed us to knockout the *MoHXT1-3* genes using a previously established methodology [20]. sgRNA complexed with Cas9-NLS was synthesized using the Guide-it sgRNA In Vitro Transcription Kit (Takara, Mountain View, CA, USA) according to the manufacturer’s instructions and purified before complexing with Cas9 using the Guide-it IVT RNA Clean-Up Kit (Takara, CA, USA). The Cas9-NLS protein was procured from Takara Bio (Inc., Kusatsu, Shiga, Japan). The complexation of Cas9-NLS with sgRNA was achieved by subjecting it to a 5-min incubation at 37 °C. The design of sgRNA was conducted online (http://www.clontech.com/). 

Donor DNA was prepared by amplifying the 1-kb sequence of the upstream and downstream homologous arms of *MoHXT2* and *MoHXT3*. Subsequently, an insert containing a selectable marker, specifically the HPH-hygromycin phosphotransferase gene cassette, was incorporated near the 5′ end of the coding regions of these genes (Appendix A). The Cas9-NLS-sgRNA ribonucleoprotein complex (RNP) was combined with the resulting donor DNA and subsequently introduced into protoplasts of the wild type A60 via transformation as the method described as Talbot et al. [21]. The coding regions of these genes facilitate the repair of double-strand breaks (DSBs) through homologous recombination with donor DNAs that possessed homologous regions on both sides of the selectable marker gene. The confirmation of the deletion was achieved through diagnostic PCR analysis (Appendix A). The positive transformants Δ*Mohxt2* and Δ*Mohxt3* were selected from a selective agar medium that was supplemented with 300 µg/mL of hygromycin B.

### 2.3. Complementation Assay of MoHXTs Mutants

The CDS region of *MoHXT2/MoHXT3* was ligated to the RP27-eGFP plasmid using *Xho* I and *Hind* III restriction enzymes, respectively. The resulting RP27-*MoHXT2* and RP27-*MoHXT3* expression constructs were amplified using primers listed in Appendix A and inserted into a pYF11 plasmid containing the bleomycin (BLM) resistance screening marker. The resulting replacement vectors, pYF11::*MoHXT2* and pYF11::*MoHXT3*, were subsequently transformed and identified following the method described by Talbot et al. [21].

### 2.4. Yeast Mutant Complementation Experiment

For the heterologous expression assays in *Saccharomyces cerevisiae*, pDR195, pDR195-*MoHXT2* and pDR195-*MoHXT3* were transformed into the hexose transporter-deficient yeast strain *EBY.VW4000* [8] using the lithium acetate method. The transformants were selected for tryptophan prototrophy on an SC medium supplemented with tryptophan and 2% maltose (SC-Trp) as the sole carbon source, without uracil, at 30 °C for 4 d. The genomic DNA of single colonies was isolated as described using the Fungal Genomic DNA Extraction Kit (Takara, Dalian, China), and the specific open reading frames (ORFs) were PCR-amplified using specific primers (Appendix A). Single transformed colonies were analyzed for their ability to grow on SC-Trp medium supplemented with 2% glucose.

### 2.5. Analysis of Conidial Morphology, Conidial Germination and Appressoria Formation

Conidia were obtained from cultures cultivated on RDC media, which was prepared by boiling 100 g of rice stalk in 1 L of sterile water for 20 min, followed by the addition of 40 g of dissolved corn flour and autoclaving at 121 °C for 20 min. Mycelium was scraped and incubated in liquid CM for 24 h. The morphological features of the mycelium were observed using SEM. To assess conidial germination and appressoria formation, the conidia were incubated on hydrophobic microscope cover glasses at a temperature of 25 °C in the absence of light. The germination of conidia and the formation of appressoria were examined after incubation periods of 0, 12, 24 and 36 h [22].

### 2.6. Pathogenicity Assay

The rice cultivar used for the pathogenicity assays was CO39. Conidial suspensions of 1 × 10^5^ conidia/mL in sterile water were sprayed on the leaves of two-week-old seedlings. The inoculated plants were then incubated in the dark for 24 h and then transferred into constant light and incubated for 5 d to assess pathogenicity [23].

### 2.7. Phylogenetic Analysis, Domain Architecture and Molecular Docking

*MoHXT* sequences were acquired from the NCBI database (www.ncbi.nlm.nih.gov) utilizing the BLAST algorithm. The construction of the phylogenetic tree was accomplished through the application of MEGA7’s neighbor-joining (NJ) method, with 1000 iterations. 

The template crystal structures were identified using BLAST and downloaded from the RCSB Protein Data Bank (PDB ID: 4LDS for MGG_00040). Homology modeling was conducted using MOE (Molecular Operating Environment) version 2016.08. To optimize the protonation state of the protein and the positioning of hydrogens, LigX was employed at a pH of 7 and a temperature of 300 K. The target sequence was aligned with the template sequence, resulting in the construction of ten distinct intermediate models. These models were generated through the permutation-based selection of various loop candidates and side chain rotamers. The intermediate model that achieved the highest score based on the GB/VI scoring function was selected as the final model. Subsequently, this model underwent additional energy minimization utilizing the AMBER12/EHT force field.

### 2.8. Expression of MoHXT2/MoHXT3 in Xenopus oocytes

Electrophysiological experiments were performed as previously described [24]. The ORFs of *MoHXT2* and *MoHXT3* were amplified using the gene-specific primer pairs listed in Appendix A. The pT7TSHA plasmid was linearized by *Spe* I/*Sph* I restriction and then subjected to recombination with *MoHXT2* and *MoHXT3* gene fragments using an In-Fusion HD Cloning Kit (Clontech, Mountain View, CA, USA). The resulting plasmid was linearized by restriction with *EcoR* I, after which the capped mRNA was reverse transcribed in vitro using the mMessage mMachine kit (www.thermofisher.com/cn/zh/home/brands/ambion.html (accessed on 26 March 2024)). *Xenopus laevis* oocyte isolation and cRNA injected were performed as previously described [24]. As the start of recording, the oocytes were subsequently bathed in modified sodium Kulori media (90 mM NaCl, 1 mM KCl, 1 mM CaCl_2_, 1 mM MgCl_2_ and 10 mM MES) at pH 5.0 with continuous perfusion at 3 mL min^−1^. The recording pipettes were filled with 3 M KCl, delivering an electrical resistance between 0.5 to 1 MΩ. The currents were measured using a Model OC-725C oocyte clamp amplifier (Warner Instruments, Hamden, CT, USA), filtered at 200 Hz and digitized at 2000 Hz. The holding potential was −40 mV, and voltage pulses from −140 to +40 mV were applied over 300 ms. Substrate-dependent currents were obtained by subtracting the average the background currents recorded before and after exposure to hexose. Baseline currents at pH 7.4 were monitored throughout the recording. The oocytes were equilibrated in the test solution for approximately 2 min before exposure to test compounds. After each exposure, the oocytes were rinsed in a hexose-free solution at pH 7.4, until the currents returned to the baseline. Data were acquired and analyzed using the software programs Digidata 1440A and pClamp10.0 (Axon Instruments Inc., Union City, CA, USA). The software program OriginPro8.0 (www.originlab.com) was used for *Km* fitting. All the experiments were performed at room temperature (22–24 °C).

### 2.9. Rapid Site-Directed Mutagenesis

Site mutation primers were designed using the Agilent QuickChange website (www.agilent.com.cn/store/primerDesignProgram.jsp (accessed on 26 March 2023)). Based on the findings from homology modeling and molecular docking, alterations were made at the 89Ala, 145Met and 421Lys sites through point mutations. The primers DTB-Ala-F/R and DTB-Ala-F/R, as provided in Appendix A, were employed. The Tiangen rapid site-directed mutagenesis kit for point mutations (Cat. No. KM101) was utilized to introduce point mutations at the corresponding three amino acid positions of the yeast vector PDR-*MoHXT2* and the *Xenopus oocyte* heterologous expression vector pT7-*MoHXT2*.

### 2.10. Transcriptome Sequencing and Analysis

The wild-type strain A60 and the Δ*Mohxt2* and Δ*Mohxt3* mutants were inoculated into liquid CM and cultured at 28 °C with shaking at 180 rpm for 50 h. Total RNA was extracted from harvested mycelia using the E.Z.N.A. Fungal RNA Kit (Omega Bio-Tek, Suite 450, Norcross, GA, USA) and RNA integrity was assessed using an Agilent 2100 Bioanalyzer (Agilent Technologies, Santa Clara, CA, USA). cDNA libraries were constructed using the TruSeq RNA Sample Preparation Kit (Illumina, San Diego, CA, USA) and sequenced on the Illumina NovaSeq 6000 platform with 150 bp paired-end (PE150) sequencing, generating at least 6 Gb of data for each sample. Quality control and filtering of raw data were performed using Trimmomatic (v0.39). Clean reads were aligned to the *M. oryzae* 70-15 reference genome (MG8, https://www.ncbi.nlm.nih.gov/assembly/GCF_000002495.2/ (accessed on 26 March 2023)) using HISAT2 (v2.1.0). Gene expression levels (FPKM values) were calculated using StringTie (v2.1.3). Differential expression analysis was performed using the DESeq2 (v1.28.1) R package, and genes with log2fold change ≥ 1 and adjusted *p*-value ≤ 0.05 were identified as differentially expressed genes. KEGG enrichment analysis was conducted using the clusterProfiler (v3.16.1) R package. The experiment was performed with three biological replicates.

## 3. Results

### 3.1. Knockout of MoHXT2 and MoHXT3, but Not MoHXT1, Affects Melanin Deposition

A total of 66 sugar transporter-like genes had been reported in the *M. oryzae* genome [18], phylogenetic analysis of which showed that they formed six clades. Four genes, *MGG_03620* (*MoST1/MoRco3*), *MGG_15700* (*MGG_01446.6/MoHXT1/MoSnf3*), *MGG_06203* (*MoHXT3*) and *MGG_00040* (*MoHXT2*) clustered in clade B with bootstrap values ranging from 91 to 94 (Appendix A). *MoST1* is required for conidiation and mycelial melanization; however, its disruption could not be complemented by the other three genes (*MoHXT1-3*) in this clade, suggesting a functional differentiation [17]. It has been reported that the expression of *MoHXT2* increases dramatically during appressorium formation, whereas that of *MoHXT1* and *MoHXT3* fluctuates, increasing 4–6 h after conidia germination and then declining, suggesting that they may play a critical role in the fungal development and pathogenicity [18]. To characterize the functions of the three *MoHXTs*, we disrupted each in the field virulent *M. oryzae* strain A60. Transformations randomly selected for correct gene replacement were verified by PCR (Appendix A).

To investigate the impact of *MoHXT* deletion on mycelial melanization, various mutant strains were cultivated on media supplemented with different carbon sources (glucose, mannose, galactose and fructose) for a duration of 5 days. The resulting colony color was then observed and compared to a control wild-type colony (A60). The findings revealed that the Δ*Mohxt2* mutants exhibited reduced melanin production compared to the control colony when grown on media supplemented with the respective hexoses. In contrast to the Δ*Mohxt2* mutants, the Δ*Mohxt3* mutants exhibited colonies that were completely white when grown on media containing one of the hexoses as the carbon source (Figure 1A shows one mutant of each gene). Through the introduction of the corresponding gene in various mutant strains, it was determined that the restoration of melanin production occurred (Appendix A). In contrast to the Δ*Mohxt2* and Δ*Mohxt3* mutants, Δ*Mohxt1* exhibited melanin deposition on all carbon source media (Appendix A). Therefore, it can be inferred that the deletion of *MoHXT1* may not impact the utilization of hexose by rice blast. Consequently, our subsequent focus was directed towards investigating the functions of *MoHXT2* and *MoHXT3*.

### 3.2. MoHXT2 and MoHXT3 Are Required for Conidiation, Appressorium Development, Pathogenicity and Cell Wall Integrity

The roles of *MoHXT2* and *MoHXT3* in conidiation were assessed. It was revealed that spare conidia were produced on the conidiophores of Δ*Mohxt3*, and only few conidia were produced in the Δ*Mohxt2* conidiophore. In comparison, the wild-type strain A60 displayed a significantly higher production of asexual conidia (Figure 1B). This finding indicates that the deletion of *Mohxt2* or *Mohxt3* has an impact on conidiation. The monitoring of conidial germination over time showed that, in contrast to the A60 control, there was a significant decrease in both spore germination and appressoria formation for Δ*Mohxt2* and Δ*Mohxt3* (Figure 1C). This observation suggests that the absence of carbon source uptake hindered the process of conidial germination.

In order to investigate the contributions of *MoHXT2* and *MoHXT3* to the pathogenicity of *M. oryzae*, the rice cultivar CO39 was subjected to inoculation with spore suspensions of A60, Δ*Mohxt2* and Δ*Mohxt3* at a concentration of 1 × 10^5^ spores mL^−1^. Subsequently, a pathogenicity analysis was conducted 96 h post-infection. Rice leaves that were subjected to inoculation with the wild-type A60 exhibited characteristic disease lesions, whereas those inoculated with the Δ*Mohxt2* strain failed to manifest disease spots, and leaves inoculated with the Δ*Mohxt3* strain only displayed minuscule disease spots (Figure 1D,E). Consequently, it can be inferred that *MoHXT2* and *MoHXT3* play a crucial role in the pathogenic growth and development of rice blast fungi.

Scanning electron microscopy (SEM) was employed to examine the mycelial cell walls of both the A60 strain and its mutants. The analysis demonstrated that the wild-type strain exhibited a fully intact tubular shape and normal morphology, whereas the mycelia of Δ*Mohxt2* and Δ*Mohxt3* mutants displayed abnormal morphology. Specifically, the cell walls of Δ*Mohxt2* mycelia appeared rough and damaged, while the surface of Δ*Mohxt3* mycelia exhibited shrinkage, local deformation and wrinkling (Figure 1F). These findings strongly suggest that *MoHXT2* and *MoHXT3* play crucial roles in the maintenance of cell wall integrity. 

### 3.3. Determination of the Substrate Affinity of MoHXT2/MoHXT3 in X. oocytes

To measure the affinity of *MoHXT2* and *MoHXT3* to each hexose, transporters were expressed in cRNA-injected *Xenopus laevis* oocytes, and a two-electrode voltage clamp was used to record hexose-dependent currents. The currents induced by hexoses were largely concentration-independent and inwardly directed and increased as the voltage became more negative. There was no observed change in the strength of the current induced by the presence of the test substrate hexoses (glucose, mannose and fructose) in the negative control (oocytes injected with water). Additionally, glucose, galactose, fructose, mannose and sucrose did not induce any current in the cell expression of *MoHXT1* (Figure 2A), which was consistent with the observation that Δ*Mohxt1* was able to produce melanin on media containing hexoses as the single carbon source. However, detectable inward currents were induced by adding hexoses to the *MoHXT2* or *MoHXT3* expression in oocytes. When the substrates were removed from the medium, the currents returned to the baseline (Figure 2B,D). These results imply that the hexose transport activities of *MoHXT2* and *MoHXT3* are proton-coupled.

A non-linear regression analysis was applied to the concentration activity plots to calculate the relevant *Km*_0.5_ values for *MoHXT2* or *MoHXT3*. The galactose-induced current was deemed negligible; thus it was excluded from further analysis. Although both *MoHXT2* and *MoHXT3* had affinities for a variety of hexoses, with the highest affinity being for glucose, a moderate affinity for mannose and the lowest affinity for fructose, *MoHXT3* showed higher affinities to the hexoses tested relative to *MoHXT2*. The current–voltage dependence curves of *MoHXT2* and *MoHXT3* on the three substrates were prepared at 5 mM. *X. laevis* oocytes were clamped at −40 mV, and the test voltage was stepped down from 40 to −140 mV in −20 mV increments over 300 ms (Figure 2C,E). 

### 3.4. MoHXT2 and MoHXT3 Restore Sugar Transport Ability in a Monosaccharide Transport-Defected Yeast Strain

To verify the functionality of *MoHXT2* and *MoHXT3*, plasmids harboring each of the genes were introduced into the *S. cerevisiae* strain *EBY.VW4000*, which is deficient in monosaccharide transport [8]. The transformed cells were cultivated on synthetic dropout (SD-Ura) media supplemented with maltose, glucose, fructose or mannose (Figure 3). Transformants were selected in a maltose liquid medium, and serial dilutions of logarithmically growing cells were made on YNB agar plates containing one of the carbon sources. Maltose was used as a positive control for growth, and the transformation carrying the empty plasmid pDR195 was used as a negative control. In the medium containing maltose, the *EBY.VW4000* strain carrying the empty vector pDR195 showed weak growth, while the strains expressing *MoHXT2* or *MoHXT3* grew well. This observation is supported by the report that yeast can absorb maltose and hydrolyze it into glucose, which can be utilized by hexose transporters [25]. It was also shown that no growth was observed on media containing hexose sugars that did not sustain the *EBY.VW4000* strain. Drop-out assays showed that the expression of *MoHXT2* and *MoHXT3* restored the growth of *EBY.VW4000* on glucose, mannose and fructose (Figure 3B). This indicates that *MoHXT2* and *MoHXT3* encode transporters that utilize multiple sugars as substrates. Overall, these results indicate that *MoHXT2* and *MoHXT3* are monosaccharide transporters. 

### 3.5. Modeling and Docking of MoHXT2 and Hexoses

Although MoHXT2 exhibited a lower affinity for hexoses than MoHXT3, it played a more important role in spore germination and pathogenicity. Therefore, MoHXT2 was selected for structural modeling and molecular docking of the hexoses (Appendix A). Ramachandran plot analysis of PepT1 revealed that a significant majority of the residues, approximately 99%, occupied the permissible regions, thus suggesting the rationality of the model’s three-dimensional (3D) structure (Appendix A). Similarly, the MoHXT2 structure exhibited conformity with the template structure. The average root mean square deviation (RMSD) value for the 3D structural alignment was determined to be 1.729 Å, and both structures displayed congruent alpha helix regions. The overall amino acid sequence identity between the two structures was found to be 31.1% (Appendix A).

The software program MOE Dock (Version 2022.02) was utilized to perform molecular docking between monosaccharides and MoHXT2 in order to predict sites playing critical roles in their binding affinity. Monosaccharides, as well as their inhibitor phlorizin dihydrate, were designated as ligands, while MoHXT2 served as the target molecule [26] (Appendix A). The 2D structures of all four molecules were transformed into 3D structures via energy minimization. The protonation state of the target molecule and the positioning of hydrogens were optimized using LigX, employing a pH of 7 and a temperature of 300 K. Prior to the docking process, the AMBER10:EHT force field and the reaction field (R-field) implicit solvation model were selected. The two template structures lacked inherent ligands, and in the absence of knowledge regarding the precise binding site, we employed the Site Finder module in MOE to forecast the pocket of MoHXT2, encompassing residues Met145, Lys421, Ala89 and Gln164. The docking procedure adhered to the “induced fit” protocol, wherein the receptor pocket’s side chains were permitted to adjust in accordance with the ligand’s conformation, while being constrained in their positions. A weight of 10 was employed to tether the side-chain atoms to their initial positions. Subsequently, the docked poses of the molecules were initially assessed based on London dG scoring, and the top 15 poses underwent force field refinement. Following this refinement, GBVI/WSA dG rescoring was conducted. The binding mode was determined by selecting the pose with the highest ranking.

The two oxygen atoms of the hydroxyl group in D-Glucose-50-99-7, regarded as hydrogen bond donors, form two hydrogen bonds with the sulfur atom of the side chain of Met145 in MoHXT2 (Appendix A). The hydroxyl group in D-Glucose-50-99-7 acts as a hydrogen bond acceptor, specifically of the nitrogen atom of the side chain of Lys421 in MoHXT2. Additionally, the hydroxyl group in D-Glucose-50-99-7 acts as a hydrogen bond donor, specifically of the oxygen atom of the backbone of Ala89 in MoHXT2. Furthermore, we conducted docking experiments of MoHXT2 with fructose, galactose and mannose (Appendix A). The results demonstrate that the residues Met145, Lys421 and Ala420 on MoHXT2 play a crucial role in binding D-Fructose, while Met145, Arg97, Ser148 and Glu149 are essential for D-Galactose binding. Additionally, Gln164, Gln299 and Gln300 are critical residues for D-Mannose recognition (Table 1). In summary, the homology model of MoHXT2 demonstrates its capability to interact with and exhibit a discernible binding energy towards glucose, fructose, galactose and mannose. 

### 3.6. Verified Amino Acid Sites Affected MoHXT2 Transport Activity 

Based on the results of the molecular docking analysis, we further validated through yeast experiments that amino acid sites Met145, Lys421 and Ala89 play a crucial regulatory role in MoHXT2 (Table 1). Three variants were constructed, namely *MoHXT2^M145K^*, *MoHXT2^K421A^* and *MoHXT2^A89R^*, each corresponding to the substitution of Met145, Lys421 and Ala89, respectively. These various were then transferred into the yeast hexose transporter deletion mutant *EBY.VW4000*. The positively transformed strains were cultivated on synthetic dropout (SD-Ura) media supplemented with different carbon sources. Transformants were selected in a maltose liquid medium, and logarithmically growing cells were subjected to serial dilutions before being plated onto YNB agar plates supplemented with glucose, fructose or mannose. Maltose was employed as the positive control for assessing growth. Our observations revealed that the *pDR::MoHXT2* control strain exhibited normal growth across various carbon sources, whereas the *MoHXT2* point mutation variant *MoHXT2^M145K^* failed to grow on hexose and exhibited impaired glucose, fructose and mannose transport. Interestingly, the growth of *MoHXT2^K421A^* and *MoHXT2^A89R^* strains was somewhat hindered in the presence of hexose-containing media (Figure 4A). These results showed that the Met145 site in the MoHXT2 transporter is crucial for hexose transport, while the impact of the K421A and A89R sites on sugar transport requires additional quantitative analysis.

To investigate this, mutations were introduced into the MoXHT2 transporter and expressed in *X. laevis* oocytes. The two-electrode voltage clamp technique was employed to measure the current response. Notably, hexose-dependent current was not detected in oocytes expressing *MoHXT2^M145K^*, suggesting that the substitution of M145 with R resulted in the loss of *MoHXT2* function. Conversely, inward currents were observed in *X. laevis* oocytes expressing *MoHXT2^K421A^* in response to varying concentrations of glucose, fructose and mannose (Figure 4B). In addition, the affinity of *MoHXT2^K421A^* towards glucose, mannose and fructose was assessed, and the current kinetics were analyzed for various substrate concentrations. The *Km*_0.5_ values for each concentration–current curve were determined using non-linear regression of the Michaelis–Menten equation. The findings indicated that *MoHXT2^K421A^* exhibited reduced affinity towards glucose and mannose compared to wild-type MoHXT2, while its affinity towards fructose remained unchanged, consistent with the results obtained from the yeast complementary test (Figure 4C).

### 3.7. Transcription Profiles of Hexose Transporters in ΔMohxt2 and ΔMohxt3 Strains

Transcriptomic analysis was conducted due to the observed differential phenotypes in the Δ*Mohxt2* and Δ*Mohxt3* strains. The A60, Δ*Mohxt2* and Δ*Mohxt3* strains were cultivated in liquid CM on a shaker for 50 h at 28 °C and 180 rpm, and their mycelia were collected for RNA sequencing. Comparison of the transcriptome data of A60, Δ*Mohxt2* and Δ*Mohxt3* strains identified 2280 differentially expressed genes, which were subjected to KEGG enrichment. In the combination of A60 and Δ*Mohxt2*, 102 differentially expressed genes were enriched in four pathways: metabolic pathways, starch and sucrose metabolism, glutathione metabolism and phenylalanine metabolism (*p* < 0.05). Notably, the starch and sucrose metabolism pathways exhibited specificity compared to other combinations. Furthermore, 195 differential genes were enriched in 12 pathways within the Δ*Mohxt3* strain, including amino acid, lipid and secondary metabolisms, in addition to the three pathways identified in the pair of A60 specimens vs. Δ*Mohxt2*. In the comparison between the Δ*Mohxt2* and Δ*Mohxt3* strains, there was an enrichment of 53 differential expression genes, with most genes enriched in the metabolic pathway (Table 2). These findings suggest that depletion of *MoHXT2* and *MoHXT3* led to changes in the metabolic pathway. *MoHXT2* influenced the primary metabolism pathway, while *MoHXT3* played more important roles in secondary metabolism. 

There are a total of 45 members of the hexose transporters protein family, and since transcription of *MGG_08617* and *MGG_10293* was not detectable, alterations in the expression levels of the remaining 43 hexose transporters in the three strains were visualized using a heatmap (Figure 5A). The findings indicated that the absence of *MoHXT2* or *MoHXT3* resulted in variations in the expression levels of genes belonging to this family, with 17 genes being upregulated and 10 genes being downregulated in both mutant strains. Conversely, the remaining 16 genes exhibited contrasting expression patterns. Specifically, 9 genes exhibited increased expression when *MoHXT2* was defective, but their expression declined when *MoHXT3* was inactivated. Conversely, 7 genes displayed opposite trends (Figure 5B). It was observed that the expression level of *MoHXT2* was significantly enhanced in the absence of *MoHXT3*, while the disruption of *MoHXT2* did not lead to an increase in *MoHXT3* transcription (Appendix A). This suggests that the deletion of *MoHXT2* resulted in an upregulation of *MoHXT3* expression as a compensatory mechanism for carbon starvation. Conversely, *MoHXT2* did not compensate for the loss of *MoHXT3* function.

## 4. Discussion

Fungi acquire an organic carbon source by utilizing sugar transporters from the major facilitator superfamily (MFS). The initial hexose transporter identified in *M. oryzea,* known as *MoST1*, is essential for conidiation and melanization, but does not play a role in pathogenicity [17]. However, understanding of the functions of other hexose transporters in the rice blast fungus is limited. Therefore, this study aims to evaluate the hexose transport capabilities of *MoHXT1-3* and explore the interaction between *MoHXT2* and *MoHXT3*. 

*MoHXT2* or *MoHXT3* is important for the full virulence of rice blast fungus. The deposition of melanin in the Δ*Mohxt2* and Δ*Mohxt3* mutants was significantly diminished, resulting in their albino phenotypes. Additionally, these mutants displayed impaired conidiation; spore germination was affected and accompanied by a loss of pathogenicity. Disruption of either *MoHXT2* or *MoHXT3* also resulted in abnormal cell walls characterized by deformed, wrinkled and damaged cells. Therefore, it is likely that the observed developmental deficits are systematic. A recent study showed that nucleotide sugar transporters (NSTs) are important for maintaining cell wall integrity and full virulence of *Magnaporthe oryzae*. These observations support the conclusion that monosaccharide starvation caused by loss of transporter function can lead to multiple physiological abnormalities, including impaired cell wall integrity, reduced melanin deposition, inhibited spore germination, metabolic disorders, etc. [27]. Despite both *MoHXT2* and *MoHXT3* being capable of promoting growth in hexose-deficient yeast strains [8], their affinities for hexoses differed in electrophysiological experiments conducted in *Xenopus* oocytes. Both *MoHXT2* and *MoHXT3* exhibited the highest affinity towards glucose, the lowest affinity towards fructose and a moderate affinity towards mannose. Glucose and fructose are, respectively, aldohexose and ketohexose. The positions of the aldehyde and carbonyl groups on the hexose molecule may greatly affect the affinity of hexose transporters for hexose, as well as absorption and transport; this may provide innovative ideas and guidance for the development of biocide molecular design targeting hexose transporters.

Despite exhibiting a diminished affinity towards the examined hexoses in comparison to *MoHXT3*, *MoHXT2* is presumed to assume more significant functions in pathogenicity. This assertion aligns with the consistent upregulation of *MoHXT2* transcription during appressorium formation, thereby bolstering the hypothesis that *MoHXT2* plays a crucial role in pathogen infection and pathogenicity. However, *MoHXT3* exhibited a notable affinity towards hexoses, as its expression escalated during conidial germination and germ-tube emergence, subsequently diminishing in subsequent infection stages. This suggests that its primary role may lie in the biotrophic phase, aiding the rice blast fungus in its competition with the host for sugar resources. Our analysis of the transcription files of the *MoHXTs* in the Δ*Mohxt2* and Δ*Mohxt3* mutants unveiled that the deletion of *MoHXT2* resulted in the downregulation of *MoHXT3*, whereas the knockout of *MoHXT3* led to an increase in *MoHXT2* transcription. These findings suggest that *MoHXT2* serves as a compensatory mechanism for the sugar starvation induced by the defect in *MoHXT3*. It may also indicate that *MoHXT2* is upstream of the *MoHXT3* pathway. It may be that *MoHXT2* regulates the function of *MoHXT3*. It is also possible that *MoHXT2* and *MoHXT3* are regulated by a common upstream transcription factor or other genes and jointly promote melanin formation and hexose transport. Both are indispensable. The KEGG enrichment analysis revealed that MoHXT2 primarily contributes to the primary metabolic pathways, whereas MoHXT3 predominantly influences the secondary metabolic pathways. 

The three genes which exhibit the highest degree of similarity to *MoST1*, namely *MoHXT1*, *MoHXT2* and *MoHXT3*, were unable to restore the functionality of the *Most1* deficient strain. *MoHXT1* demonstrates significant expression levels in both the MM medium and the infection stage in rice [18]. However, our finding indicated that *MoHXT1* was not involved in melanin formation. *Xenopus* oocytes expressing *MoHXT1* had no inward current when treated with hexose, indicating that either *MoHXT1* does not transport hexose or it requires the cooperation of a certain cofactor for full hexose transport function. It is reminiscent of the fact that *MoST1* functions as a sugar sensor that monitors carbon status around the fungal cell, especially given that it shares high sequence similarity with *CgHXT4*, a hexose transporter that shows poor sugar transport ability in *C. graminicola* [5,17]. Thus, *MoHXT1* might function as a sugar sensor, but does not fulfill a transport function. 

Based on the findings from molecular docking analysis, the amino acids Met145, Lys421 and Ala89 were subjected to point mutations in order to generate *MoHXT2* variants capable of heterologous expression in both yeast and *Xenopus oocytes*. The *MoHXT2^A89R^* and *MoHXT2^K421A^* variants exhibited compromised hexose uptake, while the *MoHXT2^M145K^* variant completely lost its ability to bind hexose in yeast. These results indicate that the Met145 site of MoHXT2 plays a crucial role in its transport function. By analyzing the interaction between it and the host, we can find the pathway and critical period of *MoHXT2* hexose absorption and transport during the infection process and focus on the differences in hexose absorption and transport genes between *M. oryzae* and host plants. Specifically designed chemical control agents inhibit the absorption and transport of hexose sugars by the rice blast fungus to achieve the purpose of inhibiting the nutrient absorption and transport of the *M. oryzae* [28], thus offering valuable insights for the development of innovative strategies to control rice blast disease by targeting carbon nutrition in pathogen–plant interactions.

## 5. Conclusions

In this study, we aim to examine the influence of three HXTs, namely MoHXT1-3, on the growth of the rice blast fungus, the infection of the host, and their affinity towards hexoses by mutants created using the CRISPR/Cas9 system. The results suggest that MoHXT1 does not participate in melanin formation or hexose transportation. Conversely, MoHXT2, despite having lower affinity for the tested hexoses compared to MoHXT3, is likely more influential in pathogenicity. Examination of the transcription profiles also revealed that the differential roles of MoHXT2 and MoHXT3 in hexose transport and sugar signal transaction. The deletion of MoHXT2 resulted in a decrease in MoHXT3 expression, whereas the knockout of MoHXT3 led to an increase in MoHXT2 transcription. Importantly, the *MoHXT2^M145K^* variant demonstrated an inability to transport hexoses.

## Figures and Tables

**Figure 1 microorganisms-12-00681-f001:**
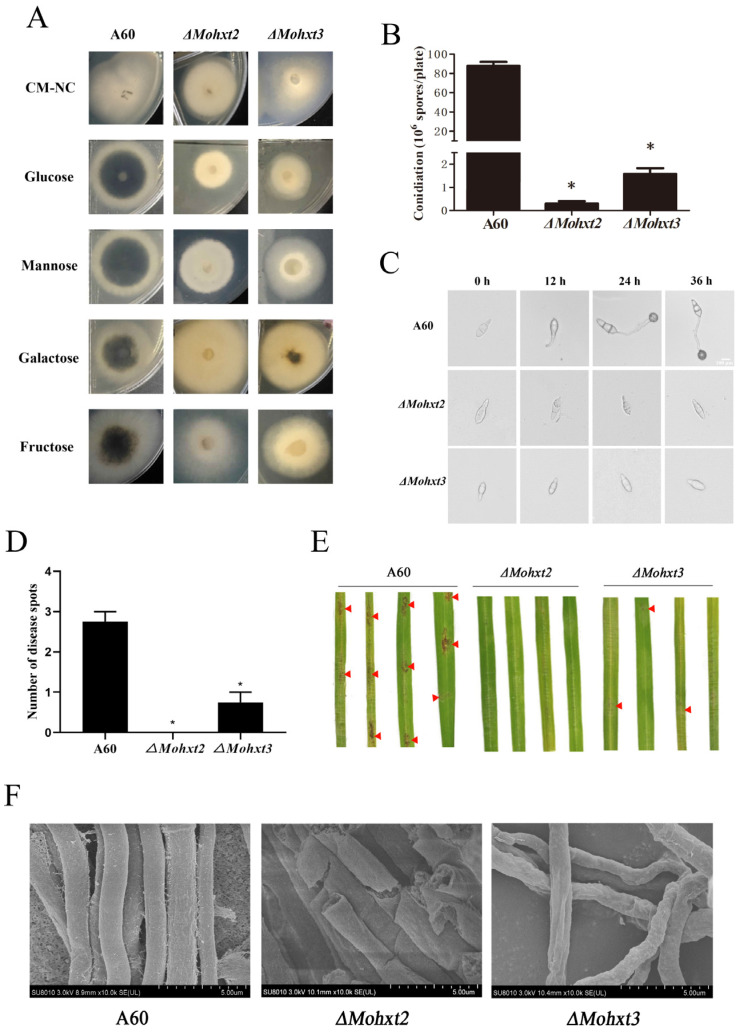
Changes to pathogenicity-associated phenotypic in Δ*Mohxt2* and Δ*Mohxt3* strains compared with the wild type A60. (**A**) Growth of the mutants Δ*Mohxt2* and Δ*Mohxt3*. Strains were cultivated on CM supplemented with various carbon sources, including glucose, mannose, galactose or fructose. A parallel control lacking any carbon source (CM-NC) was included. (**B**) Assessment of conidiation for the stains on CM plates by preparation of conidial suspension. Means are expressed as the number of conidia ×10^6^ mL^−1^ of the conidial suspension (cm^−2^) of the culture. Error bars represent standard deviations. Independent sample *t*-tests were used to determine significant differences in count data, and “*” represents significant differences between strains (*p* < 0.05). (**C**) Conidial germination ability of the wild-type and mutant strains. Drops of conidial suspension were placed on hydrophobic coverslips and kept in a moist chamber at 25 °C for 0, 12, 24 and 36 h before imaging under a light microscope. Scale bar = 100 μm. (**D**) Calculation of disease spot number produced by the strains. Independent sample t-tests were used to determine significant differences in count data and “*” represent significant differences between strains (*p* < 0.05). (**E**) Infection assays of strains on rice leaves. Droplets of conidial suspensions (1 × 10^5^ spores mL^−1^) were inoculated on 15-day-old rice seedlings (n ≥ 10) (CO39). Photographs were taken at 96 hpi. Red triangles indicate symptoms of onset. (**F**) Observation of mycelium morphological features using scanning electron microscopy (SEM). Scale bars are 5 μM for A60, Δ*Mohxt2* and Δ*Mohxt3*.

**Figure 2 microorganisms-12-00681-f002:**
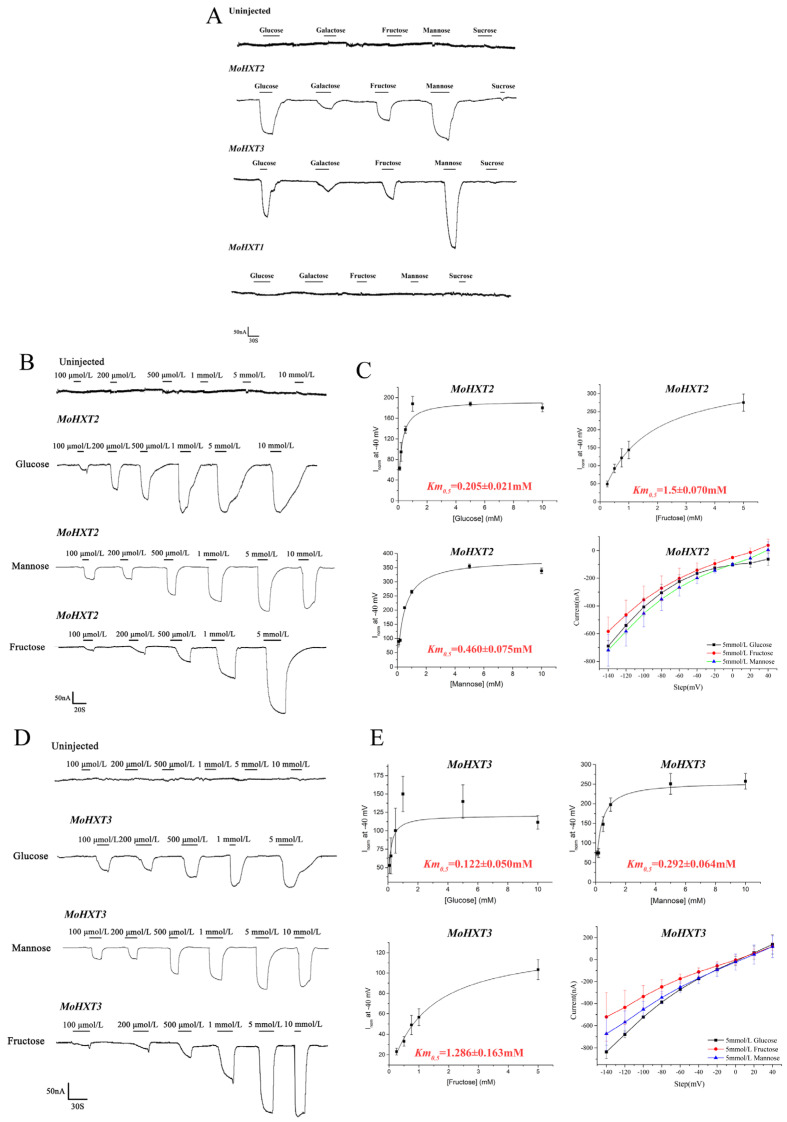
Hexose transport by *MoHXT1-3* in *Xenopus laevis* oocytes. (**A**) Inward currents induced by hexoses in *X. laevis oocytes* expressing *MoHXTs*. (**B**–**E**) Kinetic analysis of glucose, fructose and mannose transport mediated by *MoHXT2* and *MoHXT3* in *X. laevis* oocytes, where (**B**,**C**) show the fitted curve for *MoHXT2* and (**D**,**E**) show the fitted curve for *MoHXT3*. *X.* oocytes expressing the two MoHXTs were kept in Kulori solution at pH 5.0 with a clamping potential of −40 mV. Currents were normalized to Vmax. (**D**,**E**) Current–voltage relationship for *MoHXT2* and *MoHXT3* channels induced by glucose, fructose and mannose, respectively. The oocytes were clamped at −40 mV and stepped down in test voltage from 40 to −140 mV over 300 ms in −20 mV increments. The substrate-induced current (background subtraction) was measured at −140 mV. Error bars represent the mean ± SE (n = 3).

**Figure 3 microorganisms-12-00681-f003:**
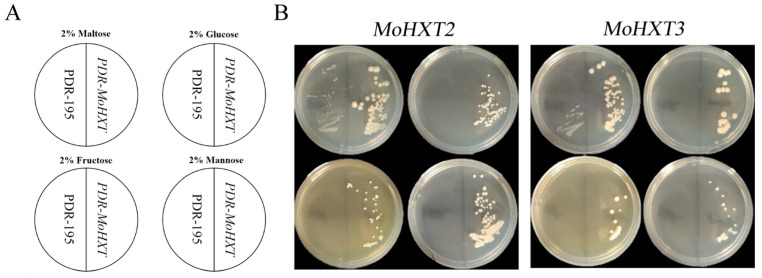
Hexose transport by *MoHXT2* and *MoHXT3* in *EBY.VW4000* yeast cells. (**A**) Schematic diagrams of relative positions for each yeast strain and the media they were grown on. (**B**) *EBY.VW4000* expression of *MoHXT2* and *MoHXT3* grown on a medium containing hexose. The growth of *EBY.VW4000* carrying the vector pDR195 was set up as a negative control in each plate, while a medium containing 2% maltose (disaccharide) was used as a positive control. Complementary yeast strains carrying pDR195-*Mohxt2* and pDR195*-Mohxt3* were spotted on media in addition to the indicated monosaccharide (glucose, mannose or fructose) and incubated at 30 °C for 144 h. These experiments were repeated three times (with three different transformants, providing three biological replicates), and similar results were obtained in each case.

**Figure 4 microorganisms-12-00681-f004:**
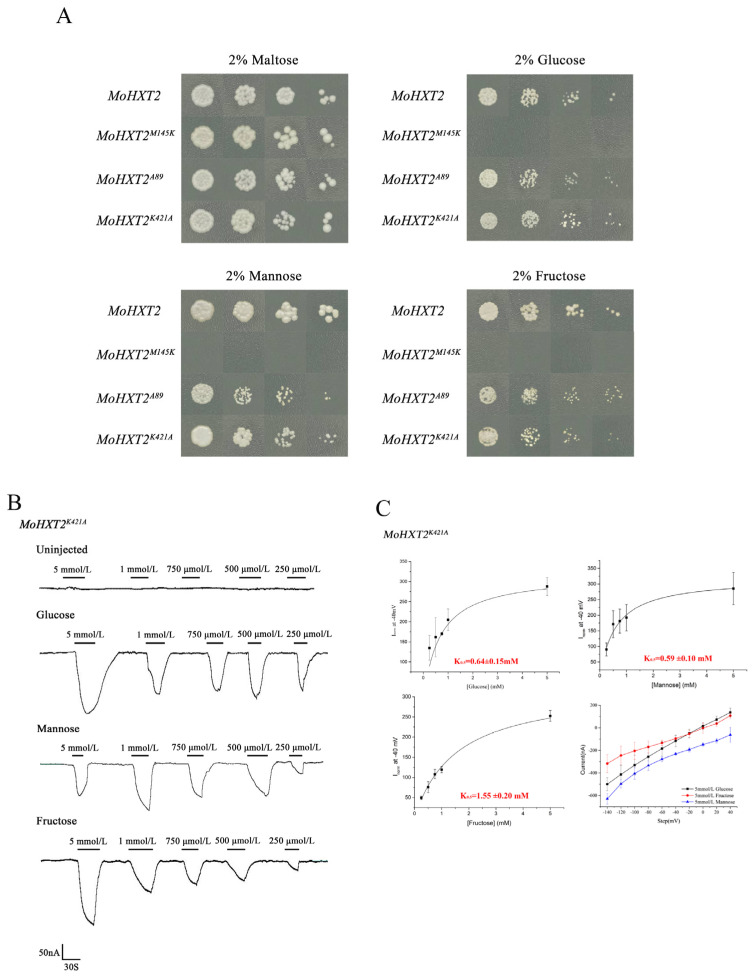
Hexose transport ability of *MoHXT2* variants. (**A**) Growth of yeast strains expressing *MoHXT2*, *MoHXT2^M145K^*, *MoHXT2^A89R^* and *MoHXT2^K421A^* on media containing 2% glucose, mannose or fructose, respectively. A positive control was employed, consisting of a medium supplemented with maltose. The yeast cell cultures were diluted in a five-fold gradient, subsequently cultivated in an incubator at a temperature of 30 °C for a duration of 3 days and then photographed. (**B**) Current–voltage reduced by different substrates in *Xenopus oocytes* expressing *MoHXT2^M145K^*. Oocytes were maintained in a Kulori media solution at pH 5.0 at a perfusion current of −40 mV and depolarized with *V_max_*. (**C**) The oocytes were clamped at −40 mV and stepped down to a test voltage between 40 and −140 mV over 300 ms in −20 mV increments. The substrate-induced current (background subtraction) was measured at −140 mV. Error bars represent the mean ± SE (n = 3).

**Figure 5 microorganisms-12-00681-f005:**
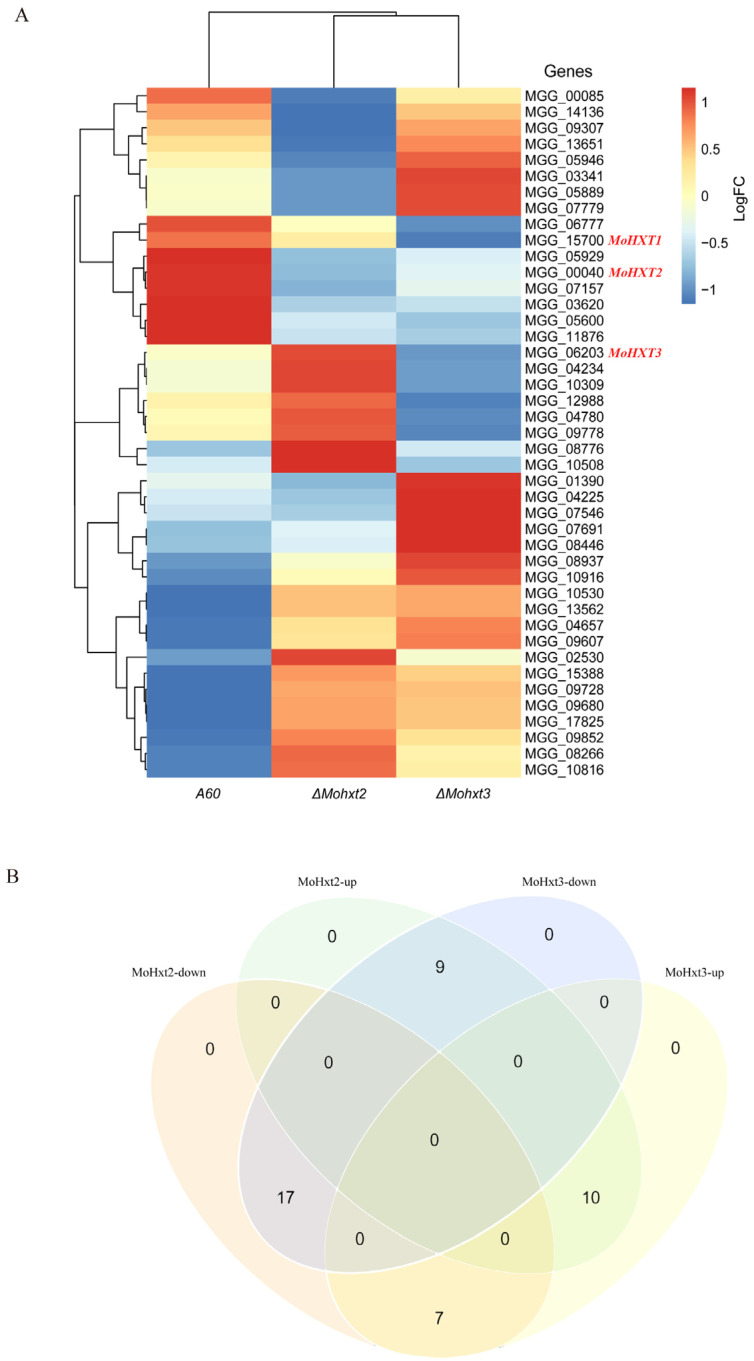
Transcription profiles of the hexose transporter-like genes in A60 and the Δ*Mohxt2* and Δ*Mohxt3* strains. (**A**) The fragments per kilobase of exon per million mapped fragments (FPKM) values for each gene are shown in Appendix A. *MGG_08617* and *MGG_10293*, which are included in the phylogenetic tree in Appendix A, are excluded from the heatmap as they were not detected in the RNA-Seq experiment. The genomes were classified based on the comparability of their expression pattern, showing the differences compared to A60. (**B**) Venn diagram of changes in genes expression levels among the three strains.

**Table 1 microorganisms-12-00681-t001:** Molecular docking between MoHXT2 and hexoses.

Receptor	Ligand	Docking Scores ^a^ (Kcal/mol)	Key Residues
MoHXT2	D-Glucose	−4.46	Met145/Ala89/Lys421
MoHXT2	D-Fructose	−5.11	Met145/Lys421/Ala420
MoHXT2	D-Galactose	−4.37	Met145/Arg97/Ser148/Glu149
MoHXT2	D-Mannose	−5.10	Gln164/Gln299/Gln300
MoHXT2	Phlorizin dihydrate	−7.02	Met145/Gln149/Arg156

^a^ Docking scores indicated the binding energy between the residues in MoHXT2 and each hexoses.

**Table 2 microorganisms-12-00681-t002:** KEGG enrichment of genes significantly differentially expressed among A60, Δ*MoHXT2* and Δ*MoHXT3* strains (*p* ≤ 0.05).

Combination of Comparisons	Enriched KEGG Terms	Number of Genes	*p* Value
A60 vs. Δ*MoHXT2*	Metabolic pathways	78	1.2 × 10^−0.5^
	Starch and sucrose metabolism	11	0.018
	Glutathione metabolism	7	0.028
	Phenylalanine metabolism	6	0.031
		Total: 102	
A60 vs. Δ*MoHXT3*	Metabolic pathways	72	4 × 10^−0.6^
	Tyrosine metabolism	12	7.1 × 10^−0.6^
	Valine, leucine and isoleucine degradation	9	0.0015
	Biosynthesis of secondary metabolites	34	0.0024
	Tryptophan metabolism	8	0.0025
	Fatty acid degradation	7	0.0025
	Amino sugar and nucleotide sugar metabolism	10	0.0047
	Glutathione metabolism	7	0.0048
	Phenvlalanine metabolism	6	0.0069
	Ubiquinone and other terpenoid–quinone biosynthesis	4	0.007
	Biosynthesis of antibiotics	23	0.026
	Glycosphingolipid biosynthesis	3	0.029
		Total: 195	
Δ*MoHXT2* vs. Δ*MoHXT3*	Metabolic pathways	43	0.00038
	Tyrosine metabolism	7	0.0028
	Taurine and hypotaurine metabolism	3	0.042
		Total: 53	

## Data Availability

The data presented in this study are available on request from the corresponding author.

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
