# Peer review of "Influence of Two Hexose Transporters on Substrate Affinity and Pathogenicity in *Magnaporthe oryzae"

_microorganisms, 2024, doi:10.3390/microorganisms12040681_

Round 1
Reviewer 1 Report
Comments and Suggestions for Authors
The authors in their study "Two hexose transporters with differential substrate affinity are important for pathognicity in Magnaporthe oryzae" investigated the impact of MoHXT1-3 on rice pathogenicity and their hexose affinity.
I have the following concerns regarding the study design
1- The 2 transporters are described as Hexose transporters, however the results revealed that MoHXT1 does not contribute to melanin formation or hexose transportation processes and MoHXT2 displays lower affinity towards the hexoses, the authors have to verify and explain this
2- Why you choose S. cerevisiae and Xenopus oocytes to evaluate their affinity for hexose (different model from Magnaporthe oryzae), again this requires further comments and whether those models have homologs of the two transporters or share the same mechanism
3- The discussion requires further improvement
Comments on the Quality of English LanguageModerate editing of English language required
Author Response
Dear Reviewer,
Thank you very much for your valuable comments We thank you for giving us an opportunity to revise and improve the quality of our article. We carefully read the comments, and we have tried our best to revise the manuscript according to the comments.
For your convenience, the text modifications are highlighted in red in the revised manuscript. We have written a point-by-point reply letter. The main purpose is to further explain the errors in the manuscript, the unclear presentation, and to correct and add some data according to your comments. In addition, some technical details, data presentation, confusing phenotype, and unclear statements have been carefully addressed in a newly revised version. We greatly appreciate your time and effort in reading our manuscript and suggesting revisions. We hope to gain your approval after careful revision. The responses to your comments are listed as below.
Response to Reviewer 1’s comments
- The 2 transporters are described as Hexose transporters, however the results revealed that MoHXT1 does not contribute to melanin formation or hexose transportation processes and MoHXT2 displays lower affinity towards the hexoses, the authors have to verify and explain this.
Response: Thank you for your valuable suggestion.
Firstly, both properties of MoHXT1 were demonstrated in the experiments. In Δmohxt1, it was found that compared with the wild type (A60), there was no melanin deficiency similar to Δmohxt2 and Δmohxt3 in different hexose carbon sources, and the growth condition was similar to that of A60. This shows that the absence of MoHXT1 does not affect the formation of melanin; we further investigated the hexose transport ability of MoHXT1 by transfecting MoHXT1 into Xenopus oocytes and expressing it, and stimulating it with different hexose, but we did not find any currents response similar to that of MoHXT2 and MoHXT3, which is sufficient to prove that MoHXT1 does not promote the formation and hexose transport of melanin formation and hexose transport.But it requires the cooperation of a certain cofactor for full hexose transport function like MoST1 and CgHXT4. MoHXT1 might function as a sugar sensor, but does not fulfill a transport function. (Line 556-577)
Secondly. Despite exhibiting a diminished affinity towards the examined hexoses in compar-ison to MoHXT3, MoHXT2 is presumed to assume more significant functions in patho-genicity. This assertion aligns with the consistent upregulation of MoHXT2 transcription during appressorium formation, thereby bolstering the hypothesis that MoHXT2 plays a crucial role in pathogen infection and pathogenicity. Our analysis of the transcription files of the MoHXTs in the ΔMohxt2 and ΔMohxt3 mutants unveiled that the deletion of MoHXT2 resulted in the down-regulation of MoHXT3, whereas the knockout of MoHXT3 led to an increase in MoHXT2 transcription. These findings suggest that MoHXT2 serves as a compensatory mechanism for the sugar starvation induced by the defect in MoHXT3. The KEGG enrichment analysis revealed that MoHXT2 primarily contributes to the primary metabolic pathways. (Line 550-560) The above reasons may be the factors affecting the difference in affinity of MoHXT2 for hexose sugars. More and more in-depth experiments may be needed to prove this conjecture.
- Why you choose S. cerevisiae and Xenopus oocytes to evaluate their affinity for hexose (different model from Magnaporthe oryzae), again this requires further comments and whether those models have homologs of the two transporters or share the same mechanism.
Response: We apologize for any confusion caused. The reason for choosing EBY.VW4000 yeast is that this yeast strain is a defective yeast strain with eighteen hexose transporters deletions, which has completely lost its ability to transport hexose. The HXT gene has high homology in Magnaporthe oryzae, and has the same transport mechanism to hexose. Using such a heterologous expression system, through the supplement experiment, the role of MoHXT2 and MoHXT3 can be observed obviously through the growth of yeast, which is a very effective and intuitive system.
The background of yeast strain comes from the following study:
Wieczorke, R.; Krampe, S.; Weierstall, T.; Freidel, K.; Hollenberg, C.P.; Boles, E. Concurrent knock-out of at least 20 transporter genes is required to block uptake of hexoses in Saccharomyces cerevisiae. Febs Lett 1999, 464, 123-128, doi:10.1016/S0014-5793(99)01698-1.
Xenopus oocytes are also an excellent model system because mature oocytes can localize many mNRAs and proteins, and also have the ability to translate exogenous microinjection RNA, which is often used for the heterologous production of various exogenous proteins. The expression system can record the potential changes on the membrane and the transport of substances simply and intuitively. MoHXT1-3 is a hexose transporters located on the membrane. Using Xenopus oocytes, the potential changes during hexose transport can be recorded very intuitively and the transport behavior can be monitored. Therefore, we will use yeast and Xenopus oocytes, two expression systems that are different from those of Magnaporthe oryzae, to conduct related research on HXT1-3.
The background of Xenopus oocytes comes from the following studies:
Lübbert, H.; Hoffman, B.J.; Snutch, T.P.; van Dyke, T.; Levine, A.J.; Hartig, P.R.; Lester, H.A.; Davidson, N. cDNA cloning of a serotonin 5-HT1C receptor by electrophysiological assays of mRNA-injected Xenopus oocytes. Proceedings of the National Academy of Sciences 1987, 84, 4332-4336, doi:10.1073/pnas.84.12.4332.
Mowry, Kimberly L. “Using the Xenopus Oocyte Toolbox.” Cold Spring Harbor protocols vol. 2020,4 095844. 1 Apr. 2020, doi:10.1101/pdb.top095844
- The discussion requires further improvement
Response: Thank you for your valuable suggestion. We have made the following additions to the Discussion section.
- Both MoHXT2 and MoHXT3 exhibited the highest affinity towards glucose, the lowest affinity towards fructose, and a moderate affinity towards mannose. Glucose and fructose are respectively aldohexose and ketohexose. The positions of the aldehyde and carbonyl groups on the hexose molecule may greatly affect the affinity of hexose transporters for hexose, as well as the absorption and transport, provide innovative ideas and guidance for the development of biocide molecular design targeting hexose transporters. (Line 547-553).
- These findings suggest that MoHXT2 serves as a compensatory mechanism for the sugar starvation induced by the defect in MoHXT3. It may also indicate that MoHXT2 is upstream of the MoHXT3 pathway. It may be that MoHXT2 regulates the function of MoHXT3. It is also possible that MoHXT2 and MoHXT3 are regulated by a common upstream transcription factor or other genes and jointly promote melanin formation and hexose transport. Both are indispensable. The KEGG enrichment analysis revealed that MoHXT2 primarily contributes to the primary metabolic pathways, whereas MoHXT3 predominantly influences the secondary metabolic pathways.(Line 565-573)
- These results indicate that the Met145 site of MoHXT2 plays a crucial role in its transport function. By analyzing the interaction between it and the host, we can find the pathway and critical period of MoHXT2 hexose absorption and transport during the infection process, and focus on the differences in hexose absorption and transport genes between M. oryzae and host plants. Specifically designed chemical control agents that inhibit the absorption and transport of hexose sugars by the rice blast fungus, to achieve the purpose of inhibiting the nutrient absorption and transport of the M. oryzae, thereby offering valuable insights for the development of innovative strategies to control rice blast disease by targeting carbon nutrition in pathogen-plant interactions.(Line 590-599)
Reviewer 2 Report
Comments and Suggestions for Authors
Reviewers comments
Two hexose transporters with differential substrate affinity are important for pathogenicity in Magnaporthe oryzae
Major comments:
- Based that the title states that hxt are important for pathogenicity, not much data supports this statement. Figure 1d: The number of disease spots in the control is rather low and not so convincing since it is unclear how many plants were inoculated. In addition, line 262 states that minuscule disease spots are observed but this is not visible in this graph. I would suggest to show pictures of the miniscule disease spots on the plant leaves. Was the small spot also counted as a disease spot?
o At this moment, it likely that the spores are not germinating when hxt2 or hxt3 are deleted resulting that no leaf disease spots are formed.
- The material and methods is lacking a lot of information that is needed to understand the setup of the experiments.
o Line 102; What does the CM contains?
o Line 109; Hxt1 was also knocked out this way?
o Missing the setup, analysis method of the transcriptomic data and the accession numbers of the data. Was this preformed in triplo/duplo? How was data treated and which software was used?
o Statistics were used but not mentioned which test was used.
o Pathogenicity test is lacking information. How many plants were inoculated (n=?)? Was a mock control done? How were the plants scored?
- Figure 1a: Why is Mohxt1 not included in figure 1a? This is important information and should be added to this figure and not the supplemental figure.
- Figure 1b and 1d: what statistical test was used?
- Figure 1c: How many spores were monitored? At this moment, the data is not sufficient to claim your conclusion that the germination rate is affected. Line 498: states that germination rates are reduced but no data is shown in this manuscript.
- Figure 2E. Mohxt3 on glucose has huge standard errors, which makes this data unreliable.
Minor comments:
- Table 2: number of genes
- Line 464: what are the genes MGG_08617 and MGG_10293, they are not mentioned before in the text. It comes out of nowhere and is rather confusing.
- Figure 5: it would be helpful if hxt1, hxt2 and hxt3 added behind the gene accession numbers.
Author Response
Dear Reviewer,
Thank you very much for your valuable comments We thank you for giving us an opportunity to revise and improve the quality of our article. We carefully read the comments, and we have tried our best to revise the manuscript according to the comments.
For your convenience, the text modifications are highlighted in red in the revised manuscript. We have written a point-by-point reply letter. The main purpose is to further explain the errors in the manuscript, the unclear presentation, and to correct and add some data according to your comments. In addition, some technical details, data presentation, confusing phenotype, and unclear statements have been carefully addressed in a newly revised version. We greatly appreciate your time and effort in reading our manuscript and suggesting revisions. We hope to gain your approval after careful revision. The responses to your comments are listed as below.
Response to Reviewer 2’s comments
Major comments:
- Based that the title states that hxt are important for pathogenicity, not much data supports this statement.
Response:Thank you very much for your reminder, we think the description of the title should be more accurate and clear, we will change it to Influence of two hexose transporters on the substrate affinity and pathogenicity in Magnaporthe oryzae
- Figure 1d: The number of disease spots in the control is rather low and not so convincing since it is unclear how many plants were inoculated. In addition, line 262 states that minuscule disease spots are observed but this is not visible in this graph. I would suggest to show pictures of the miniscule disease spots on the plant leaves. Was the small spot also counted as a disease spot?
At this moment, it likely that the spores are not germinating when hxt2 or hxt3 are deleted resulting that no leaf disease spots are formed.
Response: We are sorry for the unclear description. In Figure 1d, we inoculated more than ten rice were inoculated and obtained a relatively stable and reproducible phenotype. We have supplemented Figure 1e with pictures of tiny lesions on plant leaves. We counted small spots consistent with disease susceptibility as lesions. At the same time, we have added the legend of Figure 1 (Line 297-313).
(E) Infection assays of strains on rice leaves. Droplets of conidial suspensions (1×105 spores ml-1) were inoculated on 15-day-old rice seedlings (n≥10) (CO39). Photographs were taken at 96 hpi. Red triangles indicate symptoms of onset.
- The material and methods is lacking a lot of information that is needed to understand the setup of the experiments.
a.Line 102; What does the CM contains?
Response: We are very sorry for not providing a detailed description of the formula of CM medium. We have provided a supplementary description of the source of the CM medium formula in the method (line 105).
The CM medium we use is from the following study:
Liu, M. and Zhang, Z. (2019). Endocytosis Detection in Magnaporthe oryzae. Bio-protocol 9(15): e3322. DOI: 10.21769/BioProtoc.3322
The following is the specific formula of CM medium:
1.Liquid CM medium
10 g D-glucose
2 g peptone
1 g yeast extract
1 g casamino acid
1 ml Vitamin Solution
Hygrothermal high pressure sterilization at 121 °C for 20 min.
- Vitamin Solution
0.01 g Biotin
0.01 g Pyridoxin
0.01 g Thiamine
0.01 g Riboflavin
0.01 g p-aminobenzoic acid
0.01 g Nicotinic Acid
Add ddH2O to 100 ml and store in a dark glass bottle at 4 °C
b.Line 109; Hxt1 was also knocked out this way?
Response: I'm very sorry for the trouble this has caused you. Hxt1 was also knocked out according to the same method. We have supplemented MoHXT1 (line 112-113) in methods and the primers used in Table S4
c.Missing the setup, analysis method of the transcriptomic data and the accession numbers of the data. Was this preformed in triplo/duplo? How was data treated and which software was used?
Response: I'm very sorry for the trouble this has caused you. Regarding the software and analysis methods used for transcriptome analysis, we have provided supplementary instructions in 2.10 (Line 221-236). However, we have not yet uploaded the original data to NCBI, so we are temporarily unable to provide you with a login number.
d.Statistics were used but not mentioned which test was used.
Response: We are very sorry for not making corresponding explanations. Count data such as spore production and number of infected lesions were analyzed using independent sample t-test for significant difference analysis. Description of the data inspection has been added to the legend of figure 1b and 1d. (line 303-304 and 308-309)
e.Pathogenicity test is lacking information. How many plants were inoculated (n=?)? Was a mock control done? How were the plants scored?
Response: I'm very sorry for the trouble this has caused you. We inoculated 10 plants per treatment and performed 3 independent biological replicates. At the same time, we used water as a negative control. For the evaluation of the onset, we counted the number of lesions, which is shown in Figure 1d.(Line 307-310)
- Figure 1a: Why is Mohxt1 not included in figure 1a? This is important information and should be added to this figure and not the supplemental figure.
Response: Thank you very much for your suggestion. Mohxt1 is not included in Figure 1a for the following two reasons. Firstly,Because it was found in this study that MoHXT1 does not promote melanin formation or hexose transport. For the focus of this research paper, hexose transport Not much contribution. Secondly, because the article focuses on the contribution of MoHXT2 and MoHXT3 to sugar transport, in order to maintain the smoothness and simplicity of text writing and picture display, the phenotype of MoHXT1 is not added to Figure 1a, and is only used as a supplementary figure for supplementary explanation.
- Figure 1b and 1d: what statistical test was used?
Response: I'm very sorry for the inaccurate statement.We are very sorry for not making corresponding explanations. Count data such as spore production and number of infected lesions were analyzed using independent sample t-test for significant difference analysis. Description of the data inspection has been added to the legend of figure1band1d (line 303-304 and 308-309)
- Figure 1c: How many spores were monitored? At this moment, the data is not sufficient to claim your conclusion that the germination rate is affected. Line 498: states that germination rates are reduced but no data is shown in this manuscript.
Response: We monitored the germination status of more than 50 conidia and observed that germination was affected in all cases. Since our conclusion was inaccurate, it has been modified to state that germination was affected. (line 539)
- Figure 2E. Mohxt3 on glucose has huge standard errors, which makes this data unreliable.
Response: I'm very sorry to cause you such trouble. However, we believe that the results of Mohxt3 on glucose are credible for the following two reasons. One is because in the step experiment in Figure 2e, glucose also showed a stronger affinity, and the other is because the trend of the data points of the fitted curve also shows that the affinity of glucose is stronger than the other two sugars.
Minor comments:
- Table 2: number of genes
Response: We are very sorry for this problem. We have changed numbers of gene to number of genes in table2 and we have made modifications.
- Line 464: what are the genes MGG_08617 and MGG_10293, they are not mentioned before in the text. It comes out of nowhere and is rather confusing.
Response: We are very sorry for not clearly explaining these two genes, because in the phylogenetic tree in Figure S1, a total of 45 sugar transporters appeared, but in the RNA-seq analysis, MGG_08617 and MGG_10293 were not detected. We It is excluded from the heat map, and the corresponding description is written in the legend of Figure 5a. This approach does not easily attract the attention of readers. In order to avoid this misunderstanding, we changed it to There are a total of 45 members of the hexose transporters protein family, and since transcription of MGG_08617 and MGG_10293 was not detectable, alterations in the expression levels of the remaining 43 hexose transporters in the three strains were visualized using a heatmap (Figure 5A). (line 504-507)
- Figure 5: it would be helpful if hxt1, hxt2 and hxt3 added behind the gene accession numbers.
Response: Thank you very much for your suggestion. We have added the annotations of MoHXT1, MoHXT2 and MoHXT3 in figure 5.